# Simulation-guided Beam Search
# for Neural Combinatorial Optimization

**Jinho Choo**[*][1]  **Yeong-Dae Kwon**[*][1]  **Jihoon Kim**[1]  **Jeongwoo Jae**[1]
**André Hottung**[2]  **Kevin Tierney**[2]  **Youngjune Gwon**[1]

[1]Samsung SDS, Korea  [2]Bielefeld University, Germany

## Abstract

Neural approaches for combinatorial optimization (CO) equip a learning mechanism to discover powerful heuristics for solving complex real-world problems. While neural approaches capable of high-quality solutions in a single shot are emerging, state-of-the-art approaches are often unable to take full advantage of the solving time available to them. In contrast, hand-crafted heuristics perform highly effective search well and exploit the computation time given to them, but contain heuristics that are difficult to adapt to a dataset being solved. With the goal of providing a powerful search procedure to neural CO approaches, we propose simulation-guided beam search (SGBS), which examines candidate solutions within a fixed-width tree search that both a neural net-learned policy and a simulation (rollout) identify as promising. We further hybridize SGBS with efficient active search (EAS) [1], where SGBS enhances the quality of solutions backpropagated in EAS, and EAS improves the quality of the policy used in SGBS. We evaluate our methods on well-known CO benchmarks and show that SGBS significantly improves the quality of the solutions found under reasonable runtime assumptions.

## 1 Introduction

Combinatorial optimization (CO) problems arise in a wide variety of real-world settings in which finding high-quality solutions quickly is an important, challenging task. Efficient methods for solving CO problems have attracted significant attention, and the literature now contains vastly different styles of solution approaches. The majority relies on human-designed tricks and insights. However, neural approaches have recently begun to offer a means for data-driven optimization. Through reinforcement learning (RL), a deep neural net can automatically learn to generate heuristics that are customized to a specific context for a given dataset, without the intervention of a domain expert or the need for a pre-existing optimization approach [2].

Using a construction paradigm for neural CO methods, solutions to a CO problem can be built in a step-by-step manner following the policy calculated by the neural network. Previous work in this area tends to focus on building a good policy network model and bringing the quality of its solution generated in a single shot as close to the optimal as possible. However, research for an effective inference method for neural CO has been rare. An effective inference strategy to maximize the quality of the solution found within a given time budget is just as important, because practitioners of many industrial CO tasks are generally willing to wait even up to multiple days to find high-quality solutions to their problems.

A well-known search strategy for neural construction methods is to sample solutions from the output probability distributions calculated by the neural network (*e.g.*, [3, 4]). While this has better

36th Conference on Neural Information Processing Systems (NeurIPS 2022).

---

[*]Equal contribution. Address: jinho12.choo@samsung.com, y.d.kwon@samsung.com.

performance than single shot (greedy) solution construction, its lack of search guidance results in quickly diminishing improvements in solution quality. An alternative to sampling is Monte-Carlo tree search (MCTS) [5, 6], which completes partial solutions in the tree with rollouts. MCTS is resource intensive, potentially requiring long run times due to a multitude of rollouts and the size of the search tree. Another popular search method is beam search, which is a low-memory, heuristic version of tree search. Beam search only expands a constant number of nodes at any level of the tree search, and has been applied by several neural CO techniques [7, 8, 9, 10]. However, beam search is a greedy algorithm that blindly follows the prediction of the neural network, good or bad.

In this paper, we present simulation-guided beam search (SGBS), which combines MCTS with beam search[2] specifically for solving CO problems. SGBS performs rollouts for nodes identified as promising by the neural network, which acts as a recourse mechanism to rectify incorrect decisions by the network. Only a select number of nodes identified as promising by the rollouts are expanded, and this process repeats. The implementation of SGBS is straightforward, requiring very little modification from the basic sampling inference algorithm of any existing neural CO algorithm. Moreover, it maintains the high throughput efficiency that is the hallmark of neural network based sampling, as there is no complicated backpropagation that would prohibit batch parallelization of the search.

SGBS can be further partnered with efficient active search (EAS) [1] to achieve even better performance over longer time spans. EAS updates a small subset of model parameters at test time to "guide" the sampling towards promising regions of the search space. SGBS and EAS have a synergistic relationship when being run in alternation. The exploration-focused solution generation of SGBS can help EAS avoid local optima, while EAS provides increasingly better models for SGBS to apply during search. Thus, the contribution of this paper is two-fold: (1) we introduce the novel beam-search method SGBS, and (2) we provide a hybridized scheme using SGBS and EAS.

We show the effectiveness both of SGBS and the SGBS+EAS hybrid on a standard benchmark of the traveling salesperson problem (TSP), capacitated vehicle routing problem (CVRP), and the flexible flow shop problem (FFSP). SGBS outperforms sampling in all cases, and combining SGBS with EAS results in solutions to the TSP that have roughly two thirds the optimality gap of the previous state-of-the-art neural CO solutions, and on the CVRP roughly half the gap. On the FFSP, SGBS provides solutions roughly equivalent to the state-of-the-art in an order of magnitude less time.

## 2   Related work

**Neural combinatorial optimization**   The first application of modern deep learning methods towards a CO problem is offered by Vinyals *et al.* [7], who use their newly proposed pointer network to generate solutions to the TSP in an autoregressive manner. Their method has been improved by reinforcement learning [3] and different network architectures [11, 12, 4, 13]. Applications to other CO problems like the CVRP [8, 4] have also been proposed. More recently, the POMO method [14] has achieved state-of-the-art performance on the autoregressive construction of TSP and CVRP solutions, using a new RL method that generates diverse CO solutions that compete with each other. For these TSP-like CO problems, transformer-based neural models, which are highly specialized for fully-connected graphs, (but are more computationally expensive) are better suited and generally produce better quality solutions than the ordinary graph neural network (GNN), which are developed for universal graph topology.

Machine learning based improvement methods have also gained significant attention in the last years. Neural large neighborhood search (NLNS) [15] iteratively destroys a large part of a solution and then repairs it. Iterative local search methods [16, 17, 18, 19], however, focus more on accumulating small changes (*e.g.*, 2-opt) to a solution for improvement. The learning collaborative policies (LCP) approach [20] first generates a diverse set of candidate solutions and then improves each solution using a second policy model.

While most works focus on small to medium scale problems, large scale approaches [21, 22] have also been proposed that can solve the CVRP with up to 2000 nodes by integrating a learning method with powerful operations research solvers. Neural approaches have been made for other various

---

[2]Note that in the ML community, especially for natural language processing (NLP), the term "beam search" almost exclusively refers to beam search in which the sampling probability of each node (*i.e.* the product of all probability values encountered when moving from the root to a particular node) plays the role of the ranking function. Throughout this paper, we use "beam search" to mostly refer to this NLP-type beam search method.

types of CO problems, including routing problems [23, 24, 25], scheduling problems [16, 26, 27, 13], the satisfiability problems [28, 29], the graph problems [11, 30, 31], and electronic circuit design automation [32, 33].

**Post-hoc tree searches used by neural CO methods** Many neural CO methods use tree search or its variants (*e.g.*, beam search and MCTS) for selecting the final solution at test time. Autoregressive construction methods [7, 8] use beam search as an optional post-hoc method to boost the solution quality. Many neural approaches for routing problems rely on a predicted "heat map" that describes which edges will likely be part of an optimal solution. To generate the final solution, the heatmap is explored by, for example, a beam search [34, 9] or by MCTS [35, 36]. While strictly not considered as CO problems, many successful neural approaches on turn-based games have exploited MCTS in their inference tactics (*e.g.*, [37]).

**Other similar works** Tree search algorithms can benefit from utilizing deep neural networks in their branching and bounding mechanisms, as in learning-approaches to beam search [38, 39], MCTS [40], or branch and bound [41, 42]. MCTS variants that include a beam width, bearing resemblance to our work, have been proposed. However, unlike our method, they use random rollouts and do not require a policy for guidance [43, 44] or lack the pre-selection mechanism we propose [45]. Extensions of these variants have been successful at solving CO problems [46].

## 3 SGBS algorithm

### 3.1 Preliminaries

Consider a CO problem with $N$ decision variables $a_0, a_1, \cdots, a_{N-1}$, where one can assign $a_i$ with a value from a finite domain $X_i$ for $0 \leq i \leq N-1$. Our goal is to find a solution that maximizes the real-valued reward $\mathcal{R} : S \mapsto \mathbb{R}$. The space $S$ contains all possible solutions $s_N = (a_0, a_1, \cdots, a_{N-1})$. Constraints on the decision variables can be embedded in $\mathcal{R}$ by mapping infeasible solutions to negative infinity. In this sense, the reward function $\mathcal{R}$ directly represents the problem instances.

We construct a neural net $\pi_\theta$ parameterized by $\theta$ and use it to generate a solution one step at a time. Defining the most effective order in which to construct a solution requires a good architect, but here we assume a numerical order without loss of generality. For a partially completed ($d < N$) solution $s_d = (a_0, a_1, \cdots, a_{d-1})$, one can choose a value for $a_d$ sampled from $X_d$ following the probability distribution $\pi_\theta(a_d|s_d)$. Starting from an empty tuple $s_0 = ()$, this procedure repeats until a complete solution $s_N = (a_0, a_1, \cdots, a_{N-1})$ is generated, which we denote as $s_N \sim \pi_\theta$.

In policy-based reinforcement learning, $\pi_\theta$, $s_d$, $a_d$, and $\mathcal{R}(s_N)$ correspond to a policy, state, action, and reward, respectively. Note that the reward is given only for the final state $s_N$ in this case, and is thus equivalent to what is commonly referred to as the 'return (total cumulative rewards)' of the episode. We train $\theta$, the parameter of the policy neural network, with a reinforcement learning method such that $\theta$ is gradually updated by gradient ascent to increase the expected reward:

$$\theta \xrightarrow{\text{train by RL}} \text{argmax}_\theta \, \mathbb{E}_{\mathcal{R}' \sim P} \mathbb{E}_{s_N \sim \pi_\theta} \left[ \mathcal{R}'(s_N) \right]. \tag{1}$$

Here, $P$ refers to a (presumed) distribution of the target problem instances, from which we sample different reward functions $\mathcal{R}'$ used for training. This enables the neural net to exploit the limited size of the distribution $P$ rather than having to train for the entire problem space. As long as the given target instance $\mathcal{R}$ is indeed a part of the distribution $P$, the policy network should perform well even though it has never encountered an identical problem instance before.

### 3.2 The algorithm and its three phases

The CO problem above can be viewed as a search problem in a decision tree where each node at depth $d$ represents a partially completed solution $s_d$, and branches from this node represent different assignments to the $(d+1)$th decision variable, $a_d$. The goal now is to find a terminal node $s_N$ (or, equivalently, a path from the root to a leaf) maximizing the objective function.

The SGBS algorithm starts from the root node and builds its search tree one depth level at a time (breadth-first). At each depth level, its search procedure is carried out in three phases: expansion,

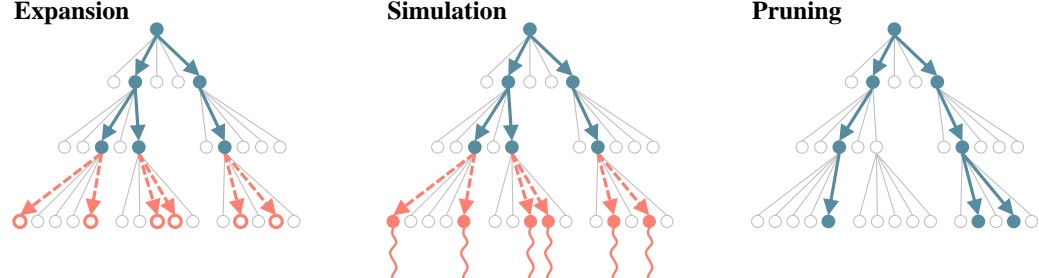

Figure 1: Three phases of SGBS, illustrated with beam width $\beta = 3$ and expansion factor $\gamma = 2$. In the 'Expansion' figure we show an example of a tree consisting of $\beta$ number of blue trajectories that represents a beam extended to depth 2. Of all its the possible child nodes, only $\gamma$ candidates (colored red) per each leaf node are selected. Wavy arrows in the 'Simulation' figure complete the rollouts, each earning a different reward value. In the 'Pruning' figure, the beam grows by another depth, keeping its width size to $\beta$ after pruning out the candidate nodes with poor performances.

simulation, and pruning (see Figure 1). It leaves at most the "beam width ($\beta$)" number of nodes for that depth level and moves on to the next. The algorithm finishes when it reaches a level in which all the selected nodes are terminal. (A terminal node expands to itself, when not all nodes in the level are terminal.) Pseudocode for the SGBS algorithm is presented in Algorithm 1.

**Expansion (pre-pruning)**   The expansion phase of SGBS is a combination of the expansion step of the regular beam search (to all child nodes) and a pre-pruning step. Given an expansion factor $\gamma$, a total of $\beta \times \gamma$ child nodes are selected, while the rest are (pre-)pruned from the search tree. That is, for each node $s_d$ contained in the beam, the top $\gamma$ of its child nodes with the largest $\pi_\theta(\cdot|s_d)$ are selected.[3] The total search time of SGBS increases (almost) linearly with $\beta$ and $\gamma$, so a user can control the balance between the performance and the speed of SGBS by adjusting these parameters.

**Simulation**   During this phase, the potential of each expanded child node is measured by a rollout, similarly to the simulation phase of MCTS. We create $\beta \times \gamma$ different CO solutions (simultaneously via parallel batch processing, if applicable) using greedy rollouts, and the rewards are tagged to the corresponding child nodes. Strictly speaking, a child node's potential can be evaluated more accurately if multiple rollouts are used (*e.g.*, repeated sampling, beam search or MCTS starting from the given node), but the use of single greedy rollout is good enough and more time efficient.

---

**Algorithm 1** Simulation-guided Beam Search (SGBS)

---

1: **procedure** SGBS(trained model $\pi_\theta$, beam width $\beta$, expansion factor $\gamma$, reward function $\mathcal{R}$)
2:    $B \leftarrow \{s_0\}$                                                                             ▷ $s_0$ is the root node
3:    **while** $\exists s_d \in B$ that is not a terminal node **do**
4:       $E, S \leftarrow \{\}, \{\}$
5:       **for** $s_d \in B$ **do**                                                                    ▷ Expansion
6:          **add** at most $\gamma$ child nodes $s_{d+1}$ with highest probabilities $\pi_\theta(\cdot|s_d)$ **to** $E$
7:       **end for**
8:       **for** $s_{d+1} \in E$ **do**                                                                 ▷ Simulation
9:          $s_N \leftarrow$ GreedyRollout($s_{d+1}, \pi_\theta$)
10:          **add** $(s_{d+1}, \mathcal{R}(s_N))$ **to** $S$
11:       **end for**
12:       $B \leftarrow$ at most $\beta$ nodes $s_{d+1}$ with highest rewards $\mathcal{R}$ in $S$         ▷ Pruning
13:    **end while**
14:    **return** $s_N$ in $B$ of highest $\mathcal{R}(s_N)$
15: **end procedure**

---

[3]As an alternative, one could group all child nodes together and use the sampling probability (accumulated from the root as is used by the NLP beam search) to select the top $\beta \times \gamma$ child nodes. This exploitation-focused approach, however, leads to worse search performance in our tests. The exploration (increased diversity from treating all nodes in the beam as equals) seems to play a more important role than the exploitation at this stage.

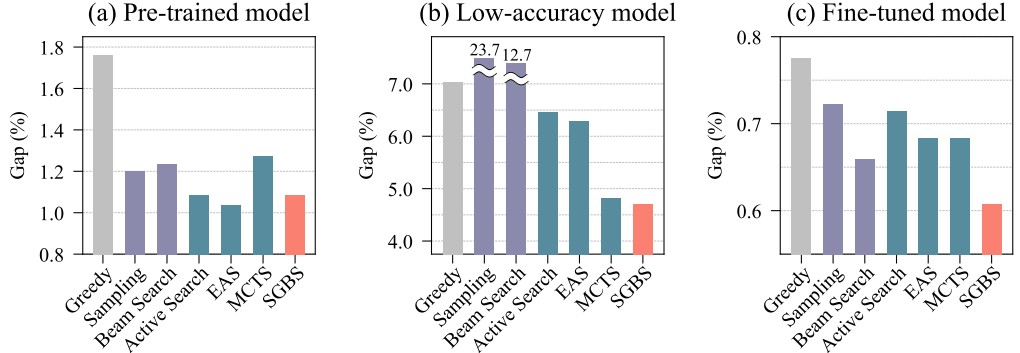

Figure 2: Performance of different search methods for the CVRP in three (a)–(c) scenarios. All methods evaluate the same number of candidate solutions per problem instance (1.2K candidates for (a) and (c), 2.4K for (b)). See Appendix C for implementation details. The gap is calculated with respect to solutions found with HGS [47, 48].

**Pruning**  We select the $\beta$ nodes with the highest simulation scores and register them as the new beam front. Note that in SGBS, no complicated backpropagation processes follow the simulation phase, keeping the algorithm simple and efficient.

### 3.3  Strengths

The implementation of the SGBS algorithm is a straightforward modification of the sampling method. It is also batch-process friendly, as the simulation phase involves parallel greedy rollouts. Despite its simplicity, SGBS can provide good search results not easily accessible to other inference methods in most settings. In the following, we discuss three different, hypothetical environments encountered by a policy network solving a CO problem and analyze how SGBS helps in each case.

**A default setting (pre-trained to a target distribution)**  Even if we assume a perfect training as prescribed in Eq. 1, the gap between the target distribution $P$ and a single problem instance $\mathcal{R}$ will cause the neural network (limited by its finite capacity) to suggest incorrect decisions from time to time. SGBS is designed to correct these mistakes on the fly. When a child node of a partial solution $s_d$ is given a relatively low score $\pi_\theta(a_d|s_d)$ by the policy network but is in fact an ideal choice, the simulation step of SGBS can prevent premature pruning of this child node.

In Figure 2 (a), we display the results of different search algorithms solving a classic benchmark CO problem, the CVRP with 100 nodes. Greedy, sampling, and beam search methods generate solutions strictly based on the probability distributions calculated by the neural network. Active search [3], EAS, MCTS, and SGBS are adaptive search methods that react to their past search results. All search methods use the output of an identical policy network and, for proper comparison, each method (except for greedy) returns its incumbent solution after investigating the same number (1.2K) of candidate solutions. We find that SGBS ($\beta = 4$, $\gamma = 4$) shows high search efficiency, outperforming most other inference methods, except for EAS[4].

**A model with low accuracy (distribution shifts)**  The power of SGBS to find good solutions really stands out when the policy network suffers from low accuracy. Dropping accuracy on a model's prediction is a common issue for many industry CO applications, where a sharp change in the distribution of the target problems are expected regularly. Without an inference method with good generalizability, these models have to be re-trained frequently, which is difficult and expensive.

In Figure 2 (b), we illustrate this case by trying to solve the CVRP with $n = 200$ instances using the neural model from (a), which was trained with $n = 100$ instances only. Now with the model

---

[4]Hottung *et al.* [1] provide three types of EAS methods in their paper: EAS-Emb, EAS-Lay, and EAS-Tab. For the sake of simplicity, throughout this paper we only use EAS-Lay and refer to it as just "EAS." EAS-Lay is the easiest to implement, and it generally shows better performance among the three, especially for the CVRP.

having relatively low accuracy, we find that sampling and beam search methods output solutions that are even worse than the greedy one (a surprising result, in fact), exposing their vulnerability on distribution shifts. The adaptive search methods (Active Search, EAS, MCTS and SGBS), however, work well, with SGBS performing the best. Note that while MCTS shows similar search quality as SGBS, it is difficult to implement MCTS to utilize the parallel computing capacity of the GPU, making it very slow (∼hours) compared to other methods (∼minutes) in our implementations.

**Fine-tuned model (trapped at a local maximum)**   The quality of the CO solution can be enhanced at test time [3, 1] by fine-tuning the parameters of the neural network to the target instance:

$$\text{pre-trained } \theta \xrightarrow{\text{fine-tuning}} \text{argmax}_\theta \, \mathbb{E}_{s_N \sim \pi_\theta} \left[ \mathcal{R}(s_N) \right], \qquad (2)$$

where $\mathcal{R}$ is specific to each target instance. Figure 2 (c) shows the results of the same search procedures used in (a), with the only change being that the neural network is fine-tuned toward each target problem before conducting the search. As expected, all search methods now produce much better solutions than they do without fine-tuning. The fine-tuned model, however, is overconfident in its first choices, and the good exploratory behavior of the model is lost as it is now trapped in a local optimum. SGBS is less affected by this confidence calibration issue [49] as its exploration range is firmly secured by the parameters $\beta$ and $\gamma$. This also explains why the hybridization of SGBS with EAS (described in the next section) leads to such a significant performance boost.

## 4   SGBS + EAS

Despite being a very efficient inference method for neural CO tasks in the short run, the capability of SGBS in making the full use of a given time budget is rather limited. First of all, SGBS is deterministic, so running it more than once offers no benefit. Its hyperparameters $\beta$ and $\gamma$ can be increased, but the gain quickly becomes marginal (see Appendix D). In this section, we introduce the second neural CO inference method, a combination of SGBS and EAS, which can fully exploit the given computation time while still utilizing the powerful search mechanism of SGBS.

EAS adjusts a small set of new parameters $\psi$ in extra layers inserted into the original neural net (with the pre-trained parameters $\theta$) using a loss function consisting of two components. The first, $J_{RL}$, aims at reducing the expected costs of the generated solutions (as in active search [3]) while the second loss, $J_{IL}$, encourages the model to output the incumbent solution with higher probability. We combine SGBS with EAS by running both methods in alternation (see Algorithm 2) with the incumbent solution and the model parameters being shared between both methods. SGBS is often able to improve the incumbent solution, helping EAS escape local optima while EAS constantly updates the model parameters, allowing SGBS to keep on growing new, non-overlapping search trees moving towards the more promising areas of the search space.

Within the inner loop of Algorithm 2, SGBS (line 5) represents the majority of the computation, occupying more than 75% of the overall runtime in most of our SGBS+EAS experiments. However, we find that the enhanced search guidance from SGBS solutions is more than enough to compensate for this. Figure 3 plots the averaged costs of the incumbent solutions found by EAS and SGBS+EAS over their runtime solving $10,000$ instances of CVRP with $n = 100$. During the 30 hours plotted in the figure, EAS and SGBS+EAS update the policy networks 600 and 150 times, respectively, for each problem instance. In spite of the large difference in the backpropagation counts, the quality of the solution more than doubles when EAS gets help from SGBS.

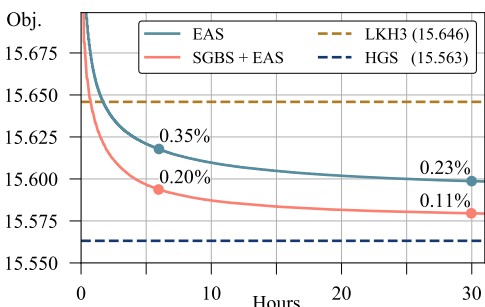

Figure 3: Performance of SGBS+EAS and EAS on $10,000$ instances of CVRP with $n = 100$, compared to LKH3 and HGS results. Percentage gaps are displayed with respect to the HGS result at the runtimes of 6 and 30 hours.

**Saving time on pre-training**   The parameters $\theta$ and $\psi$ of the policy network $\tilde{\pi}_{\theta,\psi}$ used by SGBS+EAS are trained in a two-step process: (1) $\theta$ is "pre-trained" before the target problem is presented, using the known (or estimated) problem distribution (Eq. 1). (2) $\psi$ is "fine-tuned" at test

time, working with the given target instance (Algorithm 2). One might expect that pre-training $\theta$ as much as possible before fine-tuning $\psi$ would lead to the best results. To our surprise, however, an over-trained $\theta$ leads to a worse performing SGBS+EAS, invoking the need for an early stop (see Appendix I). All SGBS+EAS experiments presented in this paper use pre-trained models with such early stops applied. This calls for interesting follow-up research topics, such as incorporating entropy regularization to enhance SGBS+EAS.

We also find the implication of the reduced dependency on the pre-training stage noteworthy. One major drawback of using a deep learning approach for real-world CO tasks is its need for constant re-training of the base model whenever a large domain shift occurs. Regardless of how short the inference time is, a neural CO approach is not viable if pre-training takes too long and cannot adapt to a changing environment. However, combining SGBS and EAS makes intensive pre-training of the policy network less crucial. For example, it takes only *two hours to pre-train $\theta$ from scratch* to prepare a neural model for SGBS+EAS, which then produces solutions for the CVRP with $n = 100$ within a reasonable runtime and similar quality to those of LKH3 (see Appendix H).

## 5    Experiments

We evaluate SGBS on various sizes of the TSP, CVRP, and FFSP instance sets with and without EAS. Our experiments are carried out on A100 GPUs (Nvidia) with 80 GB memory. We report the adjusted runtime under the premise that a single GPU has been used for each experiment.

**TSP & CVRP**    Table 1 presents the SGBS(+EAS) results on two NP-hard routing problems, the TSP and CVRP. These two benchmarks are considered the most popular in the neural CO literature. In the TSP, given a instance consisting of $n$ nodes (cities), one needs to find the shortest tour visiting every node exactly once. In the CVRP, the nodes are assigned an integer demand that must be delivered from a depot (starting node). The goal is to find the shortest-length tours that, together, visit all nodes exactly once and the sum of the demands on each tour do not exceed the capacity of the vehicle. For our experiments, we use $n = 100$ with 10,000 instances from Kool *et al.* [4]. We also perform generalization (domain shift) experiments using $n = 150, 200$ test sets of 1,000 instances from Hottung *et al.* [1]. We further explore the CVRP on a real-world like dataset in Appendix A.

We have followed Concorde [50] and HGS [47, 48] as the baseline for calculating the performance gaps in our TSP and CVRP experiments, respectively. Both are highly optimized, handcrafted solvers that produce (near-)optimal solutions to each problem type. We also compare with the results of LKH3, a well-known heuristic solver for hard routing problems. These non-neural solvers are CPU-bound and run on a single CPU only, so the wall-clock times we report in the table are not meant for direct comparison with GPU-computed numbers. Our results are compared against those of latest neural methods as well: DACT [19], NLNS [15], and DPDP [9].

We apply the SGBS methods on Transformer-like policy networks (attention model [4]) that have been pre-trained by the POMO [14] RL technique (MIT license). We perform ablation tests using the

---

**Algorithm 2** SGBS+EAS

1:  **procedure** SGBS+EAS(pre-trained model $\pi_\theta$, reward function $\mathcal{R}$)
2:      $s_N^* \leftarrow$ greedy solution generated from $\pi_\theta$                    ▷ $s_N^*$ is the incumbent solution
3:      $\tilde{\pi}_{\theta,\psi} \leftarrow$ combine $\pi_\theta$ and $L_\psi$ with randomly initialized $\psi$    ▷ $L_\psi$ is insertion layers for EAS
4:      **repeat**
5:          $s_N^0 \leftarrow$ SGBS($\tilde{\pi}_{\theta,\psi}, \beta, \gamma, \mathcal{R}$)                            ▷ Algorithm 1
6:          $s_N^i \sim \tilde{\pi}_{\theta,\psi}$ for $i = 1, 2, ..., M$
7:          $s_N^* \leftarrow s_N$ in $\{s_N^*, s_N^0, s_N^1, ..., s_N^M\}$ of highest $\mathcal{R}(s_N)$
8:          $\nabla_\psi J_{RL}(\psi) \leftarrow \frac{1}{M} \sum_{i=1}^M \left[ (\mathcal{R}(s_N^i) - b_\circ) \sum_{d=0}^{N-1} \nabla_\psi \log \tilde{\pi}_{\theta,\psi}(a_d^i | s_d^i) \right]$
                                                                                ▷ $b_\circ$ is a baseline for REINFORCE
9:          $\nabla_\psi J_{IL}(\psi) \leftarrow \sum_{d=0}^{N-1} \nabla_\psi \log \tilde{\pi}_{\theta,\psi}(a_d^* | s_d^*)$
10:        $\psi \leftarrow \psi + \alpha \left[ \nabla_\psi J_{RL}(\psi) + \lambda \cdot \nabla_\psi J_{IL}(\psi) \right]$    ▷ Policy gradient ascent
11:    **until** $\psi$ has converged or timeout
12:    **return** $s_N^*$
13: **end procedure**

---

Table 1: Experiment results on routing problems (TSP & CVRP)

| | Method | Test (10K instances) n=100 | | | Generalization (1K instances) n=150 | | | n=200 | | |
|---|---|---|---|---|---|---|---|---|---|---|
| | | Obj. | Gap | Time | Obj. | Gap | Time | Obj. | Gap | Time |
| TSP | Concorde | 7.765 | - | (82m) | 9.346 | - | (17m) | 10.687 | - | (31m) |
| | LKH3 | 7.765 | 0.000% | (8h) | 9.346 | 0.000% | (99m) | 10.687 | 0.000% | (3h) |
| | DACT [19] | 7.771 | 0.089% | (8h) | | | | | | |
| | DPDP [9] | 7.765 | 0.004% | (2h) | 9.434 | 0.937% | (44m) | 11.154 | 4.370% | (74m) |
| | POMO [14] greedy | 7.776 | 0.144% | (1m) | 9.397 | 0.544% | (<1m) | 10.843 | 1.459% | (1m) |
| | sampling | 7.771 | 0.078% | (3h) | 9.378 | 0.335% | (1h) | 10.838 | 1.417% | (3h) |
| | EAS [1] | 7.769 | 0.053% | (3h) | 9.363 | 0.172% | (1h) | 10.731 | 0.413% | (3h) |
| | | 7.768 | 0.044% | (15h) | 9.358 | 0.127% | (10h) | 10.719 | 0.302% | (30h) |
| | SGBS (10,10) | 7.769 | 0.058% | (9m) | 9.367 | 0.220% | (8m) | 10.753 | 0.619% | (14m) |
| | SGBS+EAS | 7.767 | 0.035% | (3h) | 9.359 | 0.136% | (1h) | 10.727 | 0.378% | (3h) |
| | | 7.766 | 0.024% | (15h) | 9.354 | 0.085% | (10h) | 10.708 | 0.196% | (30h) |
| CVRP | HGS | 15.563 | - | (54h) | 19.055 | - | (9h) | 21.766 | - | (17h) |
| | LKH3 | 15.646 | 0.53% | (6d) | 19.222 | 0.88% | (20h) | 22.003 | 1.09% | (25h) |
| | DACT [19] | 15.747 | 1.18% | (22h) | 19.594 | 2.83% | (16h) | 23.297 | 7.03% | (18h) |
| | NLNS [15] | 15.994 | 2.77% | (1h) | 19.962 | 4.76% | (12m) | 23.021 | 5.76% | (24m) |
| | DPDP [9] | 15.627 | 0.41% | (23h) | 19.312 | 1.35% | (5h) | 22.263 | 2.28% | (9h) |
| | POMO [14] greedy | 15.763 | 1.29% | (2m) | 19.636 | 3.05% | (1m) | 22.896 | 5.19% | (1m) |
| | sampling | 15.663 | 0.64% | (6h) | 19.478 | 2.22% | (2h) | 23.176 | 6.48% | (5h) |
| | EAS [1] | 15.618 | 0.35% | (6h) | 19.205 | 0.79% | (2h) | 22.023 | 1.18% | (5h) |
| | | 15.599 | 0.23% | (30h) | 19.157 | 0.54% | (20h) | 21.980 | 0.98% | (50h) |
| | SGBS (4,4) | 15.659 | 0.62% | (10m) | 19.426 | 1.95% | (4m) | 22.567 | 3.68% | (9m) |
| | SGBS+EAS | 15.594 | 0.20% | (6h) | 19.168 | 0.60% | (2h) | 21.988 | 1.02% | (5h) |
| | | 15.580 | 0.11% | (30h) | 19.101 | 0.24% | (20h) | 21.853 | 0.40% | (50h) |

same models for greedy solution construction, sampling, and EAS. For all POMO-based methods we employ the $\times 8$ instance augmentation technique [14] to boost the solution quality. SGBS hyper-parameters are set to $(\beta, \gamma) = (10, 10)$ and $(4, 4)$ for the TSP and CVRP, respectively. These values have been chosen based on several trial runs of SGBS+EAS with some hand-picked sets of different choices. The overall search efficiency of SGBS+EAS (per unit runtime), however, is not that sensitive to the changes of these hyperparameters as long as they are within a reasonable range (see Appendix E). We report the results of SGBS+EAS (and EAS alone) at two different points of their runtimes; one at an intermediate point and the other when the algorithm seems to have converged.

Overall, SGBS+EAS significantly outperforms all learned heuristics. On the three TSP instance sets, SGBS+EAS reduces the optimality gaps by 45%, 33%, and 35% from those of EAS, respectively. On the CVRP instances, SGBS+EAS not only outperforms EAS (52%, 56%, and 59% reduction in the gap), but also the well-known heuristic solver LKH3 by a large margin. With a gap of only 0.11% to HGS on the CVRP instances with $n = 100$, we further observe that SGBS+EAS is competitive with state-of-the-art handcrafted techniques in terms of quality, albeit not in runtime.

**FFSP** To highlight the fact that SGBS is a general inference method for any construction-type neural approaches for a CO task, not just for routing problems, we demonstrate its application on the flexible flow shop problem (FFSP), a scheduling problem that can arise (*e.g.*) in factory assembly lines. In the FFSP, jobs are processed by multiple stages in series, where each stage consists of a group of machines that perform the same task but possibly at different speeds. Each machine can handle only one job at a time so that a scheduling method is needed to decide which jobs to be processed on which machine at what time to achieve the shortest possible makespan.

We apply SGBS on the MatNet [13] based neural FFSP solver (MIT license) and experiment on the datasets of $1,000$ FFSP instances each for $n = 20$, 50 and 100 jobs. These instances [13] are based on three-stage configurations where each stage contains four different machines. The details of our SGBS experiments on the FFSP are described in Appendix G. Results of our SGBS experiments and the ablation studies are summarized in Table 2 along with the results [13] by CPLEX [51] with mixed-integer programming models and meta-heuristic solvers.

Table 2: Experiment results on $1,000$ instances of FFSP

| Method | FFSP20 | | | FFSP50 | | | FFSP100 | | |
|---|---|---|---|---|---|---|---|---|---|
| | Obj. | Gap | Time | Obj. | Gap | Time | Obj. | Gap | Time |
| CPLEX (60s) | 46.37 | 22.07 | (17h) | × | | | × | | |
| CPLEX (600s) | 36.56 | 12.26 | (167h) | | | | | | |
| Genetic Algorithm | 30.57 | 6.27 | (56h) | 56.37 | 8.02 | (128h) | 98.69 | 10.46 | (232h) |
| Particle Swarm Opt. | 29.07 | 4.77 | (104h) | 55.11 | 6.76 | (208h) | 97.32 | 9.09 | (384h) |
| MatNet [13] greedy | 25.38 | 1.08 | (3m) | 49.63 | 1.28 | (8m) | 89.70 | 1.47 | (23m) |
| sampling | 24.60 | 0.30 | (10h) | 48.78 | 0.43 | (20h) | 88.95 | 0.72 | (40h) |
| EAS | 24.60 | 0.30 | (10h) | 48.91 | 0.56 | (20h) | 88.94 | 0.71 | (40h) |
| | 24.44 | 0.14 | (50h) | 48.56 | 0.21 | (100h) | 88.57 | 0.34 | (200h) |
| SGBS (5,6) | 24.96 | 0.66 | (12m) | 49.13 | 0.78 | (47m) | 89.21 | 0.98 | (3h) |
| SGBS+EAS | 24.52 | 0.22 | (10h) | 48.60 | 0.25 | (20h) | 88.56 | 0.33 | (40h) |
| | 24.30 | - | (50h) | 48.35 | - | (100h) | 88.23 | - | (200h) |

As shown in the table, SGBS+EAS significantly outperforms the baseline traditional CO methods. While we do not claim that our approach is the state-of-the-art over all existing (neural or not) methods for the FFSP, this result shows that, at the very least, without careful design by a domain expert, non-neural approaches do not yield satisfactory solutions to many complicated CO problems. MatNet-based solvers, on the other hand, are able to produce solutions of extremely high quality, highlighting the advantage of neural CO approaches, that are automatic and purely data-driven. We have also shown that a simple change on the inference technique to SGBS+EBS empowers the existing neural solver to produce solutions that have been unattainable within a reasonable runtime.

## 6   Conclusion

We have presented Simulation-guided Beam Search (SGBS) and combined it with EAS to solve CO problems. SGBS enables neural CO methods to effectively search for high-quality solutions to CO problems and can be implemented easily in existing approaches. Furthermore, we show that combining SGBS with EAS allows for even better performance as the two techniques share information about finding solutions. Our experiments on three different CO problem settings, the TSP, CVRP and FFSP, further reduce the gap of neural CO methods to state-of-the-art handcrafted heuristics and in some cases beat methods that were state-of-the-art only a few years ago. For future work, we plan to explore setting up dynamically the parameters of SGBS and investigate the integration of SGBS and EAS more deeply.

Our code for the experiments described in the paper is publicly available at `https://github.com/yd-kwon/SGBS`.

## Acknowledgments and Disclosure of Funding

Some of the computational experiments in this work have been performed using the Bielefeld GPU Cluster. We thank the Bielefeld HPC.NRW team for their support. Furthermore, the authors gratefully acknowledge the funding of this project by computing time provided by the Paderborn Center for Parallel Computing (PC2).

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
