# Appendix

## A    Performance on real-world based instances

We further evaluate SGBS+EAS on nine real-world based instance sets from [15]. Each instance set consists of 20 instances that have similar characteristics (i.e., they have been sampled from the same underlying distribution). The instance sets differ significantly in terms of several structural properties, for example, the number of customers $n$ and their position (e.g., clustered vs. random positions). A more detailed description of instance sets can be found in [15].

One major advantage of neural combinatorial optimization approaches over traditional handcrafted optimization methods is their ability to quickly learn customized heuristics for new problem settings. With the intention to evaluate the ability of SGBS+EAS to adapt towards different problem settings, we train a different model for each of the nine instance sets. This also makes sense in light of the fact that in real-world scenarios, the characteristics of instances usually do not change unexpectedly. We train each of the nine models using the POMO method [14] for $3,000$ epochs (with $10,000$ instances each).

The operations research literature has almost exclusively focused on solving instances sequentially, even though there are real-world scenarios in which a large number of instances need to be solved in parallel. Thus, to allow for a more fair comparison to established approaches, we solve instances sequentially and not in batches. We note that this is a slightly unfavorable setting for SGBS+EAS, which has been specifically designed to exploit the parallel computing capabilities of GPUs. To account for this new evaluation setting, we always perform 10 runs in parallel for EAS and SGBS+EAS. This improves the solution quality, while leading only to a slight increase of the required runtime.

We compare the performance of SGBS+EAS to only EAS, LKH3 [52], and HGS [47, 48]. For SGBS+EAS we set $(\beta, \gamma) = (35, 5)$, the learning rate $\alpha = 0.005$ and $\lambda = 0.05$. We limit the search to $8$ iterations of SGBS+EAS, and perform 30 EAS updates in each iteration (instead of only one EAS iteration as in previous experiments). We found that performing SGBS less often, but with higher $\beta$ values improves the performance when solving instances sequentially because it better uses the available GPU memory. For EAS (without SGBS) we use identical values for $\alpha$ and $\lambda$ and we limit the runtime to the runtime of SGBS+EAS. For HGS and LKH3, we use the default parameters suggested by their developers. Note that we round the distances between customers to the nearest integer for all algorithms, as is common practice in the operations research literature. For each instance, we perform three independent runs per algorithm.

Table A shows the results. For HGS, we report the mean costs over all instances and runs. For EAS, SGBS+EAS, and LKH3 we report the absolute gap between their respective mean performance and the HGS mean. On all nine instance sets, SGBS+EAS significantly outperforms EAS. SGBS+EAS also finds solutions of higher quality than LKH3 for all nine instance sets, albeit while requiring more time to do so. Overall, SGBS+EAS almost matches the solution quality of HGS with relative gaps between 0.07% (for XE_5) to 0.83% (for XE_11). This is an impressive achievement given that

Table A: Performance for the CVRP on the XE instance sets from [15].

| Set | $n$ | HGS Costs | Absolute gap to HGS mean | | | Avg. runtime in seconds | | | |
|-----|-----|-----------|------|-------------|------|-----|-----|-------------|------|
| | | | EAS | SGBS +EAS | LKH3 | HGS | EAS | SGBS +EAS | LKH3 |
| XE_1 | 100 | 30143.7 | +63.1 | +27.1 | +641.4 | 22 | 125 | 125 | 372 |
| XE_3 | 128 | 33435.9 | +93.5 | +42.1 | +184.2 | 52 | 178 | 178 | 122 |
| XE_5 | 180 | 26561.5 | +32.3 | +17.4 | +42.9 | 44 | 341 | 341 | 65 |
| XE_7 | 199 | 28608.5 | +142.7 | +94.1 | +216.4 | 81 | 400 | 400 | 215 |
| XE_9 | 213 | 11947.2 | +78.1 | +42.3 | +131.4 | 82 | 440 | 440 | 66 |
| XE_11 | 236 | 27487.1 | +266.2 | +229.1 | +244.4 | 117 | 556 | 556 | 67 |
| XE_13 | 269 | 33949.3 | +169.6 | +139.0 | +572.3 | 103 | 921 | 920 | 343 |
| XE_15 | 279 | 44597.8 | +353.6 | +304.9 | +626.9 | 178 | 870 | 869 | 347 |
| XE_17 | 297 | 36536.4 | +205.0 | +161.0 | +495.5 | 126 | 1118 | 1117 | 152 |

HGS is a highly specialized solver, relying on handcrafted heuristics that are the result of decades of research.

## B  Implementation of greedy rollouts of SGBS

In each simulation phase of Algorithm 1, we perform $\gamma$ greedy rollouts for each node contained in the set $B$ (of size $\beta$). Among these $\gamma$ rollouts, one rollout is redundant, as it begins with the child node assigned with the largest probability. This rollout creates the same solution as one of nodes already examined in the previous simulation phase (unless we are at the root node). Therefore, in our implementation of SGBS, we improve the efficiency of our code by conducting only meaningful $\gamma-1$ rollouts. We skip the redundant rollout and simply reuse the result from the previous simulation. All in all, a total of $\beta \times (\gamma - 1)$ new candidate solutions are examined in each simulation phase.

## C  Search efficiency experiments in Section 3.3

The greedy method provides a single shot solution that represents the baseline quality of the policy network on a given problem instance. To improve the solution quality, each search method employs different tactics on how to make use of the information offered by the policy network (other than what its first choices are), as well as the evaluation results of the candidate solutions already produced. By limiting the total number of candidate solutions to the same value for all search methods, we aim to compare how effectively each search method utilizes all the information given to them.

The measurement of search efficiency in this way also reflects real-world application scenarios. While calculating the reward (or cost) of an arbitrary candidate solution is nearly instant for many benchmark CO problems used in the literature (*e.g.*, the TSP), many industry applications involve expensive reward functions $\mathcal{R}$ (*e.g.*, if the task is optimization of training hyperparameters for a neural model, the reward is its measured performance *after* the training has been executed), making it the most expensive part throughout the whole optimization process. Thus, in such cases, it is more natural to seek the best performing search method under the restriction of the allowed number of reward evaluations of candidate solutions.

**The number of candidate solutions allowed**  For experiments based on the CVRP instances with $n = 100$ (Figures 2 (a) and (c)), the number of candidate solutions allowed is 1.2K. For CVRP experiments with $n = 200$ instances (Figure 2 (b)), this number is limited to 2.4K. These limits are chosen to match the estimated total number of greedy rollouts simulated by the SGBS method when its hyperparamters $(\beta, \gamma)$ are set to $(4, 4)$.

This estimation works as follows. As explained in Section B, SGBS with $\beta = 4$, and $\gamma = 4$ makes at most $\beta \times (\gamma - 1) = 12$ new candidate solutions (greedy rollouts) at each simulation phase. A simulation phase is needed whenever it adds a customer node to a partially completed solution, which happens 100 (200) times for $n = 100$ (200). This results in 1.2K (2.4K) total greedy rollouts for each problem instance. This is actually a slightly overestimated value, because SGBS sometimes runs less than 12 rollouts during a simulation phase. This happens when there are only a few unvisited customer nodes left towards the end of the search, or when the load on the vehicle limits the number of valid next customer nodes available.

**Search details**  The sampling, active search and EAS methods can repeat the search process freely and produce as many candidate solutions as desired. Thus, we re-apply these search methods until enough candidate solutions are collected. Beam Search and MCTS, on the other hand, cannot simply be repeated. Beam Search is explicitly given a beam width of 1.2K (2.4K) from the beginning of the search, which then creates that number of complete solutions all at once. MCTS runs 12 greedy simulations at each depth level in the search tree, similar to SGBS. Our implementation of MCTS is similar to that of Xing *et al.* [35] but with their Eq.(8) changed to

$$U(s, a) = c_{\text{puct}} P(s, a) \frac{\sqrt{\sum_b N(s, b)}}{0.1 + N(s, a)}. \tag{C.1}$$

Here, $N(s, a)$ is the visit count from a node $s$ to its child $a$. The value of $U(s, a)$ decreases as more visits are made to $a$, and this encourages exploration within the MCTS algorithm. Note that $c_{\text{puct}}$ is

a constant, and $P(s, a)$ is the prior probability for choosing $a$. The constant $0.1$ in the denominator has been changed from $1$ to improve the performance of MCTS. This change induces reasonable exploration within the 12-simulation-limit, which is much less than the usual number of simulations ($> 100$) expected by the original MCTS algorithm. Finally, we do not use the $\times 8$ instance augmentation technique [14] on any of the search methods being compared in this experiment, as it would increase the number of the produced candidate solutions by a factor of 8.

**Adjustments made for POMO-trained policy network**   We use the policy neural network that is trained with RL by the POMO method [14]. The high performance of the POMO training method on the CVRP relies on the diversification of the candidate solutions induced by its manual (user-designated) selection of the first nodes. While this POMO approach leads to a significant improvement in the quality of the trained model, it results in the policy network never being properly trained for making good suggestions for the first node. This causes a complication for our search efficiency experiment, as our search methods are not provided with a proper policy for what to choose for the first customer nodes.

For the greedy results used as the baselines in Figure 2, we actually make 100 (200) different greedy solutions starting from each node, and sort them based on their performance. We use the best one to represent the greedy method. Note that this is how the inference is done in the original POMO paper. For the other search methods, however, we do not repeat the search 100 (200) times like this, as this would mean that, for example, the sampling method only gets to create 12 candidate solutions per starting node. This is wasteful, as most of the nodes are unfit to be used as the starting node. Instead, we choose $\beta = 4$ nodes as the "official" first node candidates, based on the greedy rollout results. Note that all of them are likely to be the optimal choices for starting the CVRP solution. For the sampling, active search, and EAS methods, we generate 0.3K (0.6K) candidate solutions starting from each of these four nodes. For the beam search and SGBS, we use these four nodes as the initial beam front and have the search tree expand from there. For MCTS, we simply choose just one node, which is the best out of the four, as the starting node and the MCTS method expands from there.

## D   Different sets of ($\beta$, $\gamma$) for SGBS

Figure D shows the runtime and the performance of the SGBS algorithm with different values for $\beta$ and $\gamma$ on the test set [4] of 10,000 CVRP instances with $n = 100$. The performance gap drops sharply from the greedy result as the SGBS algorithm is applied, and, with larger $\beta$ and $\gamma$, higher quality solutions are achieved. However, the performance gain quickly becomes marginal. Table 1 in the main text contains the result of ($\beta$, $\gamma$) = (4, 4).

## E   Different sets of ($\beta$, $\gamma$) for SGBS+EAS

Using a small set of instances, the performance of SGBS+EAS can be examined over different choices of $\beta$ and $\gamma$ in a relatively short time. In Figure E, we find that ($\beta$, $\gamma$) = (4, 4) gives the best

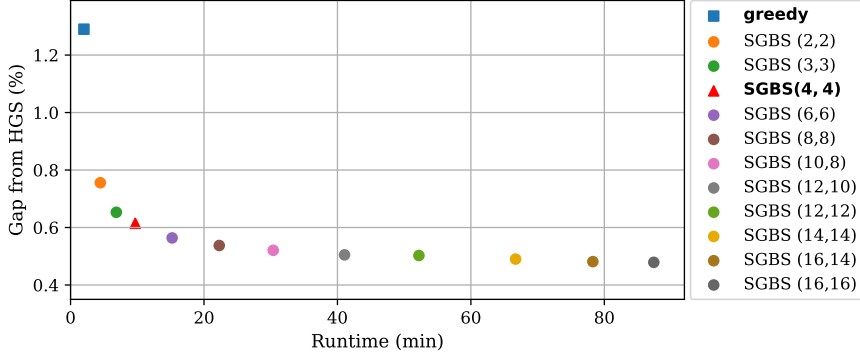

Figure D: SGBS results on the CVRP with $n = 100$, with different choices for parameters ($\beta$, $\gamma$).

result for the CVRP with 100 nodes. However, as long as the values for $\beta$ and $\gamma$ are not chosen too small or too large, the search algorithm consistently finds decent solutions. Within the range of $[2, 12]$ for both $\beta$ and $\gamma$, SGBS+EAS displays less than 0.15% gaps from the HGS solution.

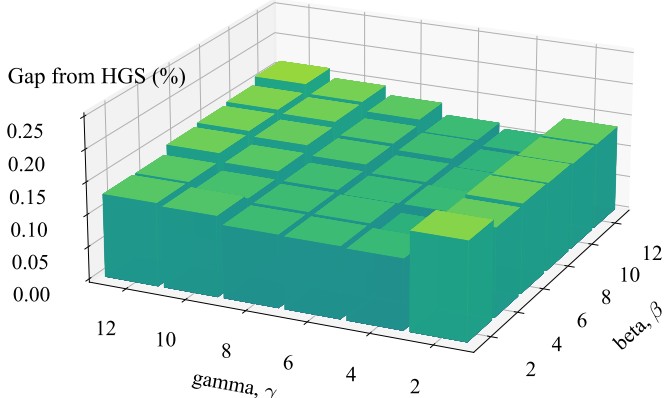

Figure E: The quality of incumbent solutions after 3-hour runs of SGBS+EAS with different values for parameters $\beta$ and $\gamma$, on 1,000 instances of the CVRP with $n = 100$.

## F   TSP & CVRP experiments

**Baselines**   We run all the baseline algorithms ourselves and compare their results and runtimes based on identical test sets in Table 1. Note that most neural baseline approaches are not explicitly designed for the generalization tests we conduct. Also, for each search method we simply re-use the hyperparameters optimized for $n = 100$ problems across all problem sizes, and therefore some generalization test results may improve substantially with appropriate adjustments to the hyperparameters. As for the DACT results on the TSP generalization tests, we opt to report no values in Table 1 as the results are not that satisfying.

**POMO model**   For our SGBS+EAS experiments and the ablation tests, we use the neural policy networks trained by the POMO method [14] ourselves. Models are regularly saved during training, and we select intermediate models that have not fully converged (*i.e.*, early stop, see Appendix I) for our experiments. The model we choose for the TSP100 experiments is trained for 1900 epochs (10 days) and for CVRP100 experiments the model is trained for 10,000 epochs (8 days).

## G   FFSP experiments

**Baselines**   CPLEX, Genetic Algorithm (GA), and Particle Swarm Optimization (PSO) results are adopted from the MatNet paper [13]. The authors have provided us the raw data so that we can display more digits for the baseline results in Table 2. The original MatNet paper reports runtimes of single-thread processes after dividing them by 8 (in an attempt to provide a more balanced view between the runtimes of CPU- and GPU-based methods). Our paper, however, uses the wall-clock time convention, so the runtimes of GA and PSO in Table 2 have been adjusted accordingly.

**MatNet model training**   For the SGBS(+EAS) experiments and the related ablation tests on the FFSP, we use MatNet architecture [13] for the policy neural network. To solve the FFSP with job count $n = 20$, 50 and 100, we train the MatNet model for 54, 95, 120 epochs (2 hours, 5 hours, 35 hours), respectively. We build and train the models based on the MatNet codes shared on the public repository, using the default hyperperameters.

**Instance augmentation**   MatNet allows different instance augmentation technique than what is used for the routing problems. By shuffling the order of the one-hot vector sequence fed to the MatNet during the initialization, one can effectively achieve different augmentations for the given

FFSP instance. Hence, for the FFSP, one can freely choose any large number as the augmentation factor. In the MatNet paper, the $\times 128$ instance augmentation is used as the default setting.

While the $\times 128$ augmentation is a reasonable choice for the greedy inference, it is too much for the SGBS(+EAS) algorithm. For better time efficiency, we use the $\times 8$ instance augmentation for all the MatNet-based search methods compared on Table 2, except for the greedy result. More specifically, we first run the greedy search using the $\times 128$ instance augmentation. Based on this result, we select the best 8 instance augmentations and use these 8 for all the other MatNet-based methods we test.

## H Short pre-tain

In Figure H, solutions found by the greedy and SGBS+EAS methods are compared to those of LKH3. A test set of 1,000 instances of CVRP with $n = 100$ is used, and SGBS+EAS is run for 3 hours. This is equivalent to a 30-hour run of SGBS+EAS on 10K problem instances, as demonstrated in Table 1. We find that the models with only a 2-hour pre-train or more can produce solutions of higher quality than those of the LKH3.

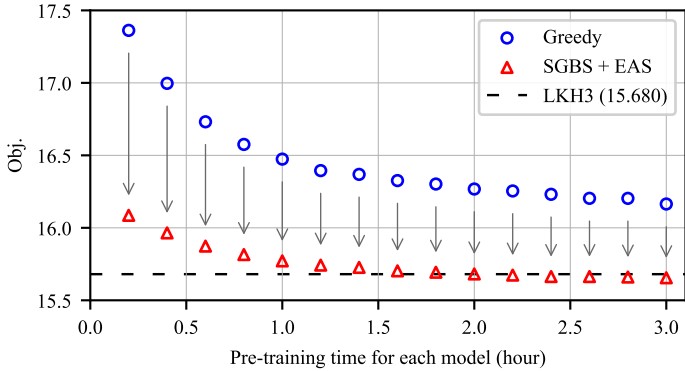

Figure H: Mean cost of the CVRP with $n = 100$ solutions produced based on different neural models, before (blue circle) and after (red triangle) applying the SGBS+EAS algorithm. Here, each model is pre-trained only for a very short time ($\leq 3$ hours) in different durations.

## I Early stops and entropy regulation

In Figure I (a) we show the mean cost of greedy solutions to the CVRP with $n = 100$, generated from a neural model trained by the POMO method. The quality of the policy network, measured by the performance of its greedy solutions, continues to improve with longer (pre-)training. While not shown in the figure, the training curve never converges completely even after 1,000 hours of training. Hence, in order to prepare a policy network to its highest quality, one is forced to wait many weeks before stopping the training.

To our surprise, however, when we use the SGBS+EAS algorithm at test time, we found that such extended pre-training of the policy neural network actually degrades the solution quality. The red triangles in Figure I (b) are the mean cost of the incumbent solutions found by a 3-hour run of SGBS+EAS. It is observed that pre-training longer than 200 hours does not help improve the solution quality, and in fact, the longer the training the worse results are returned (a positive slope of the red line). In a sense, these models are "overfit" to the distribution of the training instances, which make it more difficult to fine-tune the models to a single, specific target instance.

In order to alleviate this overfitting problem, we try regulating the entropy of the model during training. The policy gradient $\nabla_\theta J$ used by the POMO training method is described in Eq.(3) of [14], which we modify by adding the entropy regulation as

$$\nabla_\theta J' = \nabla_\theta J + \lambda_1 \frac{1}{N} \sum_{i=1}^{N} \sum_{t=2}^{M} \nabla_\theta \mathcal{H}(p_\theta(\cdot | s, a_{1:t-1}^i)). \tag{I.1}$$

Here, $\mathcal{H}$ is the entropy function. Note that the notations used in this equation follows those defined in [14], and they are slightly different from the notations used in our paper (*e.g.*, the meaning of $N$, etc.). By using a positive value for $\lambda_1$, we can increase the entropy on the output of the model. Blue squares in Figure I (b) show that this entropy regulation scheme works, and we can prevent (or at least mitigate) the overfitting problem at the pre-training stage.

We can further improve the search performance by incorporating the entropy regulation into the SGBS+EAS algorithm as well, by replacing $\nabla_\psi J_{RL}$ in line 8 of Algorithm 2 with

$$\nabla_\psi J'_{RL} = \nabla_\psi J_{RL} + \lambda_2 \frac{1}{M} \sum_{i=1}^{M} \sum_{d=0}^{N-1} \nabla_\psi \mathcal{H}\big(\tilde{\pi}_{\theta,\psi}(\cdot|s_d^i)\big). \tag{I.2}$$

Green diamond markers in Figure I (b) show that if the value of $\lambda_2$ is tuned effectively, the quality of the incumbent solutions found by SGBS+EAS can be boosted a little further.

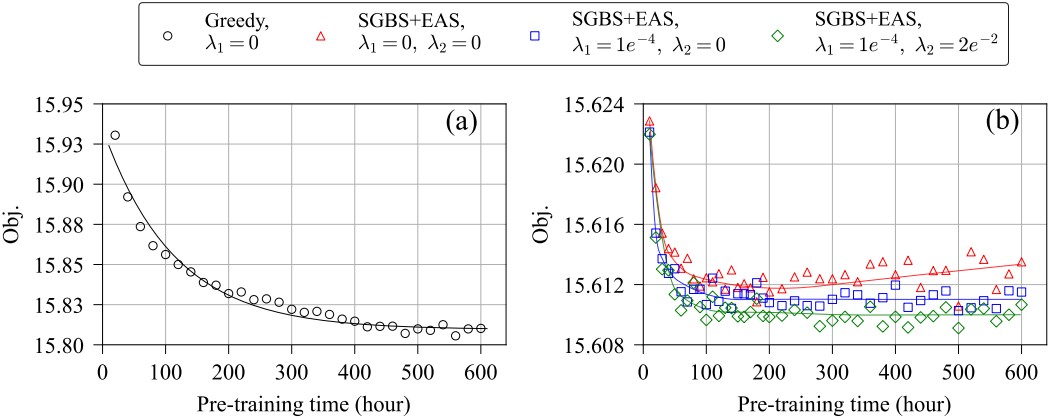

Figure I: The mean cost of the CVRP with $n = 100$ solutions found by different search methods on various models. A test set consisting of 1,000 problem instances is used. (a) shows the greedy results, and (b) shows three SGBS+EAS results with different $\lambda_1$ and $\lambda_2$ (explained in the text) after 3-hour search processes. Each data point represents a different model, pre-trained in varying degrees. Lines are drawn to guide the eye only.