# OpenReview forum: "Simulation-guided Beam Search for Neural Combinatorial Optimization"
_NeurIPS.cc/2022/Conference — NeurIPS 2022 Accept_

### Official Review · Reviewer_T9pd · 2022-07-07

**Rating:** 6
**Confidence:** 4
**Soundness:** 3 good
**Presentation:** 4 excellent
**Contribution:** 3 good

**Summary:**

This paper considers the problem of solving NP-hard combinatorial optimisation (CO) problems efficiently and near-optimally using machine learning, specifically in the context of beam search. Recent works have sought to use neural networks to evaluate and select candidates with the beam search optimisation heuristic, iteratively building partial solutions with the use of a decision tree in a breadth-first search manner until a near-optimal solution is found. However, such approaches are vulnerable to poor action evaluations made by any of the neural network’s imperfect predictions, which can be detrimental to overall performance.

To address this, the authors propose a simple algorithm, simulation-guided beam search (SGBS), which combines neural beam search with simulated rollouts similar to those used in Monte Carlo Tree Search (MCTS) algorithms but without the need for potentially complex state-action value backpropagation. Specifically, at each step in constructing the partial CO solution and the next depth-layer of the search tree, SGBS selects a group of candidate actions using a neural network to predict their complete CO solution score. It then simulates a greedy rollout from each of these actions to their terminal states, and then selects a sub-group (with size equal to the beam width) of these initially marked candidate actions to save in the search tree as the beam front, thus pruning initially marked actions with ultimately poor outcomes from the final search tree/solution. This process is repeated until a complete CO solution is found from root to leaf in the search tree.

Although SGBS alone performs well in time-restricted settings, the authors note that practitioners sometimes prefer allocating larger time budgets to find more optimal solutions. Since SGBS is a deterministic heuristic, it receives no benefit from larger time budgets other than adjustment of its hyperparameters, $\beta$ and $\gamma$, which the authors show have limited effect on performance beyond a certain point (which is quickly reached). To address this, the authors combine SGBS with the recently published ‘efficient active search’ (EAS) method of Hottung et al. 2022 to create a new algorithm, SGBS-EAS. The authors show that SGBS and EAS have a symbiotic relationship with one another whereby SGBS helps EAS to avoid local optimal while EAS provides an increasingly performant neural network model during test-time. Both SGBS and SGBS-EAS are shown to outperform some canonical CO heuristics and state-of-the-art ML solvers in both solution quality and solving time.

**Questions:**

* **Context and related work:** The proposed SGBS and SGBS-EAS algorithms, as well as the baseline agents compared to, rely on relatively expensive (in terms of both training time and inference) transformer models with additional training overhead at test time and/or simulated rollouts. Can the authors comment on how such techniques compare to other state-of-the-art ML-CO methods based on cheaper graph neural network models which do not require simulated rollouts (Dai et al. 2017, Abe et al. 2019, Li et al. 2018, Barrett et al. 2020 and 2022, Drori et al. 2020)? Would there be differences in optimality and training and inference time performance? Might the training overhead of EAS at test time incur disadvantages at test time in terms of both inference time and solution efficacy when initially deployed? How does SGBS/SGBS-EAS fit in the context of these other literature contributions?

* **Reward function:** In the SGBS Methodology section, the authors state that $P$ represents a set of target problem instances from which they sample different ‘reward functions’ $R’(S_N)$ during training, which enables the neural network to exploit the limited size of the distribution $P$ rather than having to train for the entire problem space. I’m not entire sure what this means - is it that $S_N$ (i.e. the final solution found by the RL agent) is changing and therefore that the total return evaluated by the reward function is changing (since the RL agent explores during training), or are the authors actually using different reward functions to evaluate the same solution $S_N$ in different ways? If the latter, what reward functions are used? How can the RL agent learn to predict the values of states and actions when the reward function in the MDP is changing? I do not entirely follow this MDP formulation or which reward function(s) were used for each CO problem class.

* **Action sampling methodology:** What policy is used for selecting actions in the SGBS greedy rollouts? Is it just the argmax of the $\pi_\theta(\cdot | s_{d})$ policy of the neural network, or some other heuristic?


### Miscellaneous minor issues
* Fig 1: Would be useful to have a key showing what the different colours, fills, and shapes of search tree edges and nodes mean. Also it may be helpful to add in the caption that Fig 1 is an SGBS step at $d=2$ for clarity

* Pg 7 line 218: ‘the’ typo: ‘...are displayed with the respect to...’

* Pg 4 line 129: Should ‘...the rewards are tagged to the...’ be ‘...the returns are tagged to the...’ given standard RL jargon which refers to per-step evaluation signals as rewards and total cumulative rewards across the episode as returns?


### References

* Andre Hottung, Yeong-Dae Kwon, and Kevin Tierney. Efficient active search for combinatorial optimization problems. International Conference on Learning Representations, 2022.
* Hanjun Dai, Elias B. Khalil, Yuyu Zhang, Bistra Dilkina, and Le Song. Learning combinatorial optimization algorithms over graphs. In Advances in Neural Information Processing Systems, 2017
* Kenshin Abe, Zijian Xu, Issei Sato, and Masashi Sugiyama. Solving NP-Hard Problems on Graphs by Reinforcement Learning without Domain Knowledge. arXiv:1905.11623, 2019
* Zhuwen Li, Qifeng Chen, and Vladlen Koltun. Combinatorial Optimization with Graph Convolutional Networks and Guided Tree Search. In Advances in Neural Information Processing Systems, 2018
* Thomas Barrett, William Clements, Jakob Foerster, and Alex Lvovsky. Exploratory combinatorial optimization with reinforcement learning. In Proceedings of the AAAI Conference on Artificial Intelligence, 2020
* Thomas D. Barrett, Christopher W. F. Parsonson, and Alexandre Laterre. Learning to solve combinatorial graph partitioning problems via efficient exploration. arXiv:2205.14105, 2022
* Iddo Drori, Anant Kharkar, William R. Sickinger, Brandon Kates, Qiang Ma, Suwen Ge, Eden Dolev, Brenda Dietrich, David P. Williamson, and Madeleine Udell. Learning to solve combinatorial optimization problems on real-world graphs in linear time. arXiv:2006.03750, 2020

**Strengths And Weaknesses:**

Strong points:
* All sections of the paper are excellently written and easy to understand.
* As far as I am aware, the idea to combine neural beam search and simulated rollouts in this way is novel, even if it is simple.
* The proposed method is easy to implement and integrate with existing neural beam search techniques.
* The experimental results outperform baselines on standard NP-hard CO problems; an important and significant application area of ML.

Weak points:
* Lack of comparison to/discussion of some other state-of-the-art graph neural network techniques which do not require simulated rollouts or search trees (see below).
* A few things which could be clarified (see below).

---

> ### Author Response · Authors · 2022-08-02
> **Response to Reviewer T9pd (2/2)**
>
> &nbsp;
>
> **Q2. Reward function**
>
> We understand your confusion, and we will refine the texts in the section in our final version of the paper. The confusion comes from the fact that we are using the term 'reward function' and 'problem instance' interchangeably. Why this is so is explained in line 104 and its proceeding lines. In our notation, $s_N$ is just a set of values assigned to decision variables, that exist independently from the details (parameters and constraints) of a specific CO problem. We first choose what problem instance to solve and only then we can evaluate how good a solution $s_N$ is by applying the reward function $\mathcal R$ (defined by the parameters and constraints of the problem instance) on $s_N$. There is an one-to-one correspondence between $\mathcal R$ and the problem instance in this notation, hence the interchangeability.
>
> The use of the symbol $\mathcal R$ to represent a problem instance was a way to make our formal mathematical formulations concise, but it is true that it can sometimes be misleading.
> We hope that our explanations above have resolved your confusion. When we say that we sample different reward functions, we simply mean that we are sampling the training data (random CO problems instances) for our neural network.
>
> &nbsp;
>
> **Q3. Action sampling methodology**
>
>
> You are right. The greedy rollouts simply follow the argmax of the policy neural net's outputs.
>
> &nbsp;
>
> **Minor issues**
>
> Thank you for pointing out the typo (line 218) and the ambiguous use of the term 'reward' (line 139 and other places), and for your helpful comments regarding Fig 1. We will reflect them on the revised version of our paper.

---

> > ### Comment · Reviewer_T9pd · 2022-08-06
> > **Reviewer T9pd response**
> >
> > Most of the points in my review have received an excellent response - thank you for your clarifications and for addressing the concerns I had.
> >
> > On the point of the model comparisons - I am aware that a transformer can be thought of as a fully connected GNN, but performing computations on a fully connected graph of course brings additional computational costs, which will presumably impede training and solving times to some extent. The authors have addressed the transformer vs. GNN differences in terms of optimality, but not training + inference time.
> >
> > I would also still like to know whether the authors think that the training overhead of EAS at test time may incur disadvantages at test time in terms of both inference time and solution efficacy when initially deployed (inference time because you must perform network updates at test time, and solution efficacy because initially no test time training will have taken place and therefore performance will presumably be poorer).
> >
> > Will the authors be updating their paper with the relevant ML4CO context, related work, further work/limitations with respect to GNNs for different problem types etc. before the end of the discussion period?

---

> > > ### Author Response · Authors · 2022-08-08
> > > **Thank you for your feedback**
> > >
> > > &nbsp;
> > >
> > > **Transformer vs. GNN**
> > >
> > > Training time (i.e., time to train a neural net to properly encode graphs) depends on many factors, the model architecture being just one of them. In our experience with Transformers, training times differ widely on different problems even if we use the exact same model (due to the different ranges of the target instances, the different aspects of the graph features to learn, etc.). As the Transformer and the GNN have been applied mostly on two non-overlapping subsets of the CO problems, direct comparison in their training times can be misleading. Even for cases like the TSP, where both models have been tried, the vast difference in their performances indicates that the two models are learning different features of the graphs.
> > >
> > > However, it is safe to say that training a GNN is much more time-efficient than training a Transformer in general, especially when the graphs that need to be encoded are sparse. The same can be said for the inference times as well, although in this case the inference time would mostly depend on the inference strategy itself rather than the model type.
> > >
> > > Speaking of training times, an example of the training curves for our Transformer-type model is shown in Appendix I, Figure I(a). For the CVRP with 100 nodes, it takes a few weeks to (pre-)train our model, assuming that a single GPU is used.
> > >
> > >
> > > &nbsp;
> > >
> > > **Overhead of EAS at test time**
> > >
> > > EAS does not incur any meaningful overhead at test time, because it does not require any change to the (pre-trained) policy model for it to be applied. In other words, at the early stage of EAS, when the effect of EAS training is negligible, the policy model is simply in its pre-trained state. The solutions produced by the EAS method at this point have the same quality as those without EAS (i.e., those of the sampling method). The rate at which the solutions are produced goes down a little because EAS needs to perform extra computations to change the parameters of the model, but this makes only a marginal change in the overall speed. EAS affects only the outer layer of the model and the backpropagation is quick (compared to the original Active Search, which does the full-scale backpropagation).
> > >
> > >
> > > &nbsp;
> > >
> > > **Paper Revision**
> > >
> > > In the first paragraph of the Related work section, we will add a few sentences to compare Transformer models and the GNN models based on the contents of our discussion here.
> > >
> > > In the third paragraph, we will add references to the GNN approaches as the examples of the NCO techniques for the graph CO problem (as there should have been). And we will also explicitly mention that SGBS(+EAS) can be applied to some of these existing neural approaches to other types of CO problems.
> > >
> > > We are not sure about updating the paper yet. We do not get the one extra page for the revision during this discussion period. We also plan to revise our paper reflecting comments from the other reviewers as well.

---

> ### Author Response · Authors · 2022-08-02
> **Response to Reviewer T9pd (1/2)**
>
> Thank you for reviewing our paper. We find your review helpful and insightful. Let us answer to your questions below.
>
> &nbsp;
>
> **Q1. Context and related work**
>
> &nbsp;
>
> ***- Neural Approaches on Graph CO problems***
>
> Thank you for pointing out the important branch of neural combinatorial optimization, the graph CO problems using the neural ML approach. Honestly, we had them in our related work section while we were preparing our scripts, but somehow they had gone missing in the midst of frenzy process of putting everything together right before submission. We are grateful that you have noticed, and we will include them in the related work section in our revision. Graph neural net (GNN) based CO methods, such as those found in the references you have provided, are particularly effective on solving graph CO problems and thus have potential in making significant impacts in the real world. The Maximum Cut problem, for example, is a fundamental CO problem to which many NP-complete CO problems found in the industry can be reduced.
>
> &nbsp;
>
> ***- Model Comparisons***
>
> In order to position our work properly in the context of these popular GNN approaches, we must first point out that, technically speaking, the transformer-like models we use in our paper can be also considered as a type of GNN. These models, however, differ from regular GNN models in that they are highly specialized for fully-connected graphs. As such, they fit nicely to the types of CO problems we focus in our paper (TSP, CVRP, FFSP, etc.), but they cannot be directly implemented for many graph CO problems dealing with various topological graph structures.
>
> Ordinary GNN models, as they can be applied to graph problems of any topology, have been employed in various ways to solve the aforementioned fully-connected-type CO problems, such as TSP. These attempts, however,  have not led to quite satisfactory results yet. For example, Dai et al. 2017 solves TSP with 100 nodes with roughly ***7%*** gap to the optimal solution, and Drori et al. 2020 demonstrated ***3%*** optimality gap for the same problem. This is to be contrasted with ***0.1%*** optimality gap of the POMO model we use for TSP100 in our paper. (And the POMO model achieves this level of solution quality in just single greedy rollouts. EAS-SGBS reduces this optimality gap down to ***0.02%***.) GNN models, developed for universal graph topology, seem to have hard time competing with specialized transformer-like models on these fully-connected-type CO problems.
>
> Applications of neural net models in the other way around, on the other hand, i.e. adapting transformer models to graph CO problems, have not been explored as extensively as should have been by the ML community. We believe this is a very interesting and possibly quite promising future research topic.
>
> &nbsp;
>
> ***- Construction vs. Improvement***
>
> Most learning-based methods for CO problems in modern literature can be categorized into either the construction-type or the improvement-type. A construction-type method relies on a neural network to provide a policy for building a high quality solution from ground zero. Choices made during the construction are irreversible, making the method difficult to perform on complex and highly unpredictable environments (i.e. CO problems). An improvement-type method, on the other hand, usually starts with randomly generated, low quality solutions, but keeps on modifying them into better ones. This provides a more flexible and consistent approach that usually scales betters.
>
> Both methods have been explored on solving graph CO problems. Dai et al. 2017, Li et al. 2018 and Drori et al. 2020 are construction-type approaches, whereas Barrett et al. 2020 and 2022 are improvement-types. SGBS(+EBS) algorithm we introduce in our paper is an improvement method, but it is very unique in that it is designed to be implemented on top of the pre-existing construction-type methods. This means that our method can be easily integrated with existing construction-type graph problem solvers, such as those in Dai et al. 2017 and Drori et al. 2020, and can improve their performances even more.

---

### Official Review · Reviewer_QRWC · 2022-07-11

**Rating:** 7
**Confidence:** 2
**Soundness:** 3 good
**Presentation:** 4 excellent
**Contribution:** 3 good

**Summary:**

This paper introduces a general approach for solving combinatorial optimization problems using a hybrid Beam-Search approach that can update it's policy on the fly. In contrast to work that focuses on improving policy towards getting solutions in a single shot, this work attempts to make better use of available computation.  One trick here is replacing the compute-intensive MTCS with a simpler Beam-search to pick rollout candidates. Additionally, the authors use EAS to update the policy network to make more effective choices at test time. This paper contribution is in domains where queries to a ground truth simulator are cheap but the search space is combinatorial.

**Questions:**

.

**Limitations:**

I'm limited in my ability to evaluate this paper, especially in the broader context. I read the paper, and understand the main points, but I cannot place it in the wider literature.

I'm confident it is well executed, and sensible in it's approach.

**Strengths And Weaknesses:**

### Strengths
- Well-motivated paper.
- Clean execution.
- Code is provided and readable.
- Algorithm is tested on reasonable benchmarks.

- I appreciated the section on the algorithm's strengths.


### Weaknesses
Overall I found it a good paper, no major flaws.

---

> ### Author Response · Authors · 2022-08-02
> **Response to Reviewer QRWC**
>
> Thank you and we appreciate your nice review of our work. They are accurate and clearly summarized.

---

### Official Review · Reviewer_q7i4 · 2022-07-12

**Rating:** 4
**Confidence:** 4
**Soundness:** 3 good
**Presentation:** 3 good
**Contribution:** 3 good

**Summary:**

The paper presents a Beam Search strategy augmented with Monte Carlo Rollouts evaluating the “return” potential of a sampled action, instead of the commonly used policy likelihood. Besides the immediate advantage of using a search mechanism on top of a network inference (as opposed to a single-shot inference), the paper highlights that the rollouts can act as a recourse mechanism to rectify incorrect decisions made by the network (when selecting the top-k most likely actions). To leverage the additional time allocated to the generation of a solution, the method is combined with the recently published Efficient Active Search (EAS) method. The paper shows interesting synergies between these two components. The approach and its variants are validated on a set of combinatorial optimization problems (TSP, CVRP, FFSP) as well as compared to Beam Search, EAS, MCTS and non-adaptive methods.

**Questions:**

- The rollout mechanism depends on the neural network as well — Could an argument be made against the fact that we assume that the network might make a mistake at step *t* but then takes a sensible sequence of greedy actions till the end of the episode (recourse mechanism of the SGBS)?
- Caption of Fig 2: “All methods evaluate the same number of candidate solutions per problem instance.” Can you describe what it means in terms of MCTS simulations, Gradient descents applied to the network for Active Search, etc?
- Fig 2 - What do these numbers vary as we grow or reduce the compute budget. Here the setup has been set to favour SGBS using (β = 4, γ = 4). Does the MCTS reach a higher asymptotic performance when given additional time? Given that the authors mentioned in the appendix that SGBS quickly plateaus, where (resources) do the methods cross in terms of performance? I'm surprised that a simple Beam Search with a random rollout to evaluate the nodes outperform the more structured MCTS search. Could the authors provide additional details on this?
- Fig 2c - why does the MCTS’s performance not follow the same performance trend as the SGBS when fine-tuning the model? SGBS seems to benefit more than the MCTS when the model is fine-tuned.
- I haven’t found the implementation of the MCTS or the Beam/SGBS search in the code linked to the paper. It seems that only the networks and environments are made available. Will that change in the future?

**Limitations:**

-

**Strengths And Weaknesses:**

**Originality**

The method presented is relatively simple, i.e. augmenting the beam search procedure with greedy rollouts. Although I'm not aware of the exact same scheme being published before, the originality of the method is limited. In addition, most of the performance is driven by EAS. Interesting synergies are highlighted and constitute a novel contribution. Overall, although the method is well described and investigated, the originality and contribution are slightly under expectations.

**Significance**

Machine Learning for Combinatorial optimization problems is an important area of research as this family of problem it underlies many real-life applications. I don't consider the search mechanism to be of significant importance but rather see an interest in the discussion around EAS + SGBS. The search is however a good suggestion for practitioners to easily implement a lookahead mechanism (easier and more scalable than MCTS).

**Quality:**

The paper is technically strong. The results are well reported (tables and figs), documented and discussed. The balance in the level of details provided in the main text and the appendix is adequate for the reader.

**Clarity:**

The quality of the writing meets the expected standards. The content is well structured and logically organized. The results are appropriately reported and discussed.

---

> ### Author Response · Authors · 2022-08-02
> **Response to Reviewer q7i4 (3/3)**
>
> &nbsp;
>
> **Q3&Q4. Problem with MCTS applied on RL-trained models**
>
> At the heart of MCTS is the 'selection policy,' which controls the balance between exploration and exploitation during its selection phase. (This is not to be confused with the rollout policy, the output from the policy neural network.) For example, one of the most well-known 'selection policies' is the UCT (Upper Confidence Bound 1 applied to trees) formula used for the classical random-move-based MCTS. To run MCTS not with random moves but with moves following a given policy model, one needs a 'selection policy' that can incorporate this prior knowledge. We use the most common 'selection policy' formula in the deep learning community, the one used by 'AlphaGo' [35] as well as the MCTS-TSP paper [33] that we have used as the model for our MCTS implementation. (For readers who are interested in more details of our MCTS implementation, we refer to [33] or the Python code we share in our repository.) This formula includes the term $U(s, a)$ (see Eq. C.1), which is proportional to the prior $P(s,a)$, the output from the policy neural network.
>
> Our paper focuses on policy models that construct high-quality CO solutions trained by the policy-gradient method (Eq. 1). Over the past few years, this new learning-based way of tackling complex CO problems has gained increasing popularity and progressively more successful results on various types of problems. One of the characteristics of this method is that the policy model it produces tends to show overconfident behavior, outputting disproportionately high probability values for its most favorable choices. This behavior is normal, and it is encouraged by the RL method itself because it results better outcomes. (When you are placing a bet on a game that you know you have a 60% chance to win, you want to play it 100% of the time, not just 60% of the time.)
>
> Now, for example, if one child node has a prior probability of 99% for being selected (which we see happening very frequently in our models) while all the others share the rest, the MCTS selection formula will almost never select other child nodes and run simulations for them. This is because the conventional 'selection policy' for deep learning applications we use does not have a strong enough drive for exploration that can offset such extremely imbalanced prior. This is to be contrasted with our SGBS algorithm, which will always check a pre-defined number of child nodes for their potentials with simulations regardless of how imbalanced the policy is.
>
> It may be possible to mitigate this incompatibility issue between MCTS and the policy models trained by RL. One could try inserting a very strong entropy regularization term into the loss during the model training to never allow such strong preference from the model in the first place (at the expense of the decreased solver performance). Or, one could tweak the selection formula directly with a few tunable parameters to allow reasonable exploration even in the cases like the one described above. Solving this issue within MCTS is certainly an interesting topic of its own, but it is beyond the scope of our work. In our current settings, MCTS does not outperform SGBS for most problem instances, even if we allow a much larger time budget for it (such as 1200 simulations per step, instead of 12 as used in Figure 2).
>
>
> Lastly, this incompatibility issue also explains the performance of MCTS shown in Figure 2B and Figure 2C. When the model faces unfamiliar problems (Figure 2B, low accuracy model) and is unsure what the good choices are, the problematic over-confidence disappears, allowing good explorations for MCTS to run. We find that MCTS performs almost as good as SGBS in this case. In Figure 2C, on the other hand, fine-tuning the model exacerbates the over-confidence problem, and the performance of MCTS becomes worse than a simple beam search.
>
> &nbsp;
>
> **Q5. Codes for MCTS and Beam Search**
>
> We have uploaded MCTS and Beam Search codes to the github repository we provided. You also said you cannot locate the implementations of the SGBS codes, but they are written in '*Tester.py' files, and you can run them using 'test.py' files. Please forgive our bad filename choices, and sorry for the confusion.

---

> ### Author Response · Authors · 2022-08-02
> **Response to Reviewer q7i4 (2/3)**
>
> &nbsp;
>
> **Q1. Assumption for the recourse mechanism of SGBS to work**
>
> When the network is in a state where it makes only a single mistake during a greedy rollout from producing a globally optimal solution, then SGBS will certainly correct it and return the optimal solution. In practice, of course, the network can make many mistakes along the way that SGBS rarely produces an optimal solution in a single run. (This is why we also have invented SGBS+EAS so that SGBS can be applied repeatedly.)
>
> What we find most fascinating about SGBS is not the fact that it can sometimes produce optimal solutions, but rather the fact that it consistently improves the existing solution closer to the optimal solution better than any other existing search methods, under a reasonable computational budget. This does not require an assumption such as there being just a single mistake of the model for it to work. SGBS is capable of finding an improved solution with fewer number of mistakes (or less severe mistakes) when the model makes so many more in its greedy rollout.
>
> &nbsp;
>
> **Q2. Evaluation of MCTS, Active Search, and EAS in Figure 2**
>
> The detailed explanations you seek are described in Appendix C, under the paragraphs titled with 'Search details' and 'Adjustments made for POMO-trained policy network.' We provide a simplified answer here.
>
> Take, for example, Figure 2A, for which we limit the number of candidate solutions to be created to 1.2K for each problem instance. MCTS needs to run simulations each time before choosing a node definitively, which needs to happen at least 100 times for CVRP-100. Therefore, we let MCTS run 12 simulations at each depth level of the search tree (which is the same for SGBS).
>
> Active Search and EAS use POMO-training for their gradient descents, which is the RL method that has also been used to train the neural net model in the first place. More specifically, in order to execute Active Search (or EAS), four (4) sampling rollouts are drawn, and their results are averaged to make a baseline. Using this baseline, the policy gradient descents are applied upon every probability sampled during the rollouts. The procedure is repeated 300 times that make up a total of 1.2K rollouts being produced.

---

> ### Author Response · Authors · 2022-08-02
> **Response to Reviewer q7i4 (1/3)**
>
> Thank you for such a thorough assessment of our paper. We'd like to clarify some of the points that might have been clearer. This reviewer's deeply insightful comments and questions are particularly helpful for the next revision.
>
> &nbsp;
>
> **Originality**
>
> You have summarized our SGBS algorithm as the beam search procedure augmented by greedy rollout evaluation. The term 'beam search' used here could mean two different things as we have mentioned in Footnote 1 of the paper. If you mean beam search in the most general sense of the word as in https://en.wikipedia.org/wiki/Beam_search, then your description is correct. But then this will be a very weak ground for judging our algorithm to be of small originality. If this is the case, please see our answer to Reviewer 7y6f on the comparison of SGBS with Monte-Carlo Beam Search (MCBS).
>
> If you mean beam search as the word is commonly used in the machine learning community, which we think is more likely, then we must point out that your summary is inaccurate. Previous work on construction-type RL methods has used beam search based on the policy likelihood, a common procedure used in many other areas of machine learning as well. This is certainly the most sensible and natural way of ranking a node. This policy likelihood, or the output probability of a partial solution, is calculated as the product of all probabilities encountered as the search moves from the root to a particular node. Our SGBS algorithm, however, *does not rely on this policy likelihood at all*. A variation of SGBS that does use it is described in Footnote 2 of our paper, but we have found it inferior and never have presented it formally in the paper. While we would not repeat the explanation of SGBS procedures here, we strongly argue that our SGBS algorithm is NOT just a simple extension to the common 'beam search' method, the one already being used actively by the community. Instead, it is a completely new approach toward executing beam search (in the general sense), specialized for policy networks trained to create solutions for complex CO problems.
>
> &nbsp;
>
> **Significance**
>
> We are delighted to hear that you find the combination of SGBS with EAS an interesting technique, and we hope that you would also find the SGBS algorithm a meaningful invention as well. Please check our comments on 'Originality of SGBS' above, as well as our answers to your Q3&Q4 below.
>
> In short, SGBS is not merely a simple extension to an existing technique. It is an unconventional tree search technique (to neural CO researchers), applied to the RL-trained policy networks for the very first time. As such, our work can lead to many follow-up papers, extended by other researchers interested in creating better decision-time planning methods using policy neural networks.
>
> We have shown that SGBS outperforms existing tree search methods, including MCTS. While SGBS is similar to MCTS in many conceptual ways, it should be noted that MCTS is ill-suited for working with RL-trained policy neural networks in terms of its performance, at least in the current form commonly accepted by the deep learning community.
>
> And finally, while you did indicate that you appreciated the scalability and the ease of implementation of SGBS, we would still like to invite you to read our answer to Reviewer 7y6f on 'MCTS that can replace SGBS'. There, we have described just how difficult it is to design an efficient and practical tree search method like SGBS that can be used for neural combinatorial optimization.

---

> > ### Comment · Reviewer_q7i4 · 2022-08-08
> > **Novelty of the method**
> >
> > I don't understand the authors when they mention that SGBS "does not rely on this policy likelihood at all." Algorithm 1 indicates that the most likely actions according to the trained policy $\pi$ are selected as candidates before being evaluated using greedy rollouts. Hence my comment.
> >
> > I agree, however, with footnote 2, stating that classic Beam Search usually accumulates the likelihood of the samples from the root onwards which is not done in this work. It seems indeed natural to avoid infinitely penalising the generated trajectories based on the likelihood of the sequence of actions whereas a more informative signal is available thanks to the greedy rollouts.
> >
> > Hopefully, my understanding of the method is aligned with what's described in the paper. If so, I maintain my original statement regarding the novelty of the method. I'm mindful that this is, to a degree, subjective. I consider the modifications to the Beam Search algorithm for CO problems to be below my expectations for acceptance, although the work was scientifically well conducted and presented.

---

> > > ### Author Response · Authors · 2022-08-09
> > > **Thank you for your feedback**
> > >
> > > Sorry for confusing you with our ambiguous use of the term 'policy likelihood'.
> > > Your description and understanding of the method is accurate, indeed.
> > >
> > > We value your opinion and we thank you again for your hard work and time for reviewing our work.

---

### Official Review · Reviewer_7y6f · 2022-07-15

**Rating:** 6
**Confidence:** 4
**Soundness:** 4 excellent
**Presentation:** 3 good
**Contribution:** 2 fair

**Summary:**

This paper proposes two novel methods for neural-network-based combinatorial optimization.
The two methods, SGBS and SGBS+EAS show promising performance for solving three types of combinatorial optimization problems (TSP, CVRP, and FFSP).


**Questions:**

In my understanding, in the EAS paper [1], the solutions are simply sampled from the model.
Was there any theoretical/practical difficulty in extending the idea to MCTS?

I agree that SGBS easily utilizes batch parallelism, but it is not the only way.
Have you considered comparing the proposed method with existing parallel MCTS?
In my intuition, parallel MCTS equipped with progressive widening may also perform.

---
Post rebuttal comments.

Thank you for the detailed comment.
However, I did not think I needed to change my score.

**Limitations:**

There is a discussion about the two hyperparameters $\beta$ and $\gamma$.

**Strengths And Weaknesses:**

Strengths:
The authors proposed Simulation-Guided Beam Search (SGBS), combining Monte-Carlo Tree Search (MCTS) and beam search.
SBGS is also combined with an interesting existing technique, Efficient Active Search (EAS), which updates a small part of the network during the search.
The experiments show that SGBS+EAS performs better than many existing approaches.

Weaknesses:
The novelty is somewhat limited.
The key idea of EAS is already proposed in [1].
SGBS is very similar to an existing work, Monte-Carlo Beam Search [41].
The contribution of this paper is to prove that EAS works well when combined with MCTS.

---

> ### Author Response · Authors · 2022-08-02
> **Response to Reviewer 7y6f**
>
> Thank you reviewing our paper, and we are grateful for the comments and important questions raised. We answer to your questions below, highlighting our responses to key concerns.
>
> &nbsp;
>
> **Novelty of the SGBS algorithm**
>
> We disagree with your statement that SGBS is very similar to Monte-Carlo Beam Search (MCBS*) [41] to the point that the novelty of SGBS is undermined. SGBS and MCBS are, in fact, built upon two totally different search strategies at their cores. Their superficial resemblance comes from the fact that they both rely on a single (or possibly the equal number of) rollouts from each candidate child node, the result of which determines whether it makes it to the next beam front or not. But the mechanism behind such uniform use of rollouts among the candidate nodes are vastly different between MCBS and SGBS.
>
> MCBS explicitly avoids the delicate mechanism for balancing the exploration and exploitation (compared with the use of UCT in MCTS) for simplicity's sake. It simply gives all child nodes of the beam front equal chances for playouts. (i.e., maximum exploration within the beam scope at the cost of search inefficiency.) MCBS may work well on simple abstract problems with *small branching factors*, but it is not suitable for more complicated problems (the TSP with 100 nodes, for example).
>
> SGBS, on the other hand, is more like MCTS than MCBS in carefully shaping its scope of exploration for better search efficiency. Unlike the classic MCTS that is based on random moves, however, we have equipped SGBS with a powerful policy neural network. The neural net limits the scope of exploration by making preliminary decisions on which child nodes get to have the chance for a playout, right on the spot, without having to go through many (MCTS-like) simulations. This is the expansion (pre-pruning) phase of SGBS.
>
> A tree-search method, neural or nonneural, is identified and categorized by how it deals with the problem of balancing exploration and exploitation, which is the fundamental challenge for any tree-search algorithm. A beam search strategy aided by policy neural network via pre-pruning process (not just on the rollouts) is thus novel and an extremely powerful concept that we are introducing to the community.
>
> -----
> [\*] MCBS can be constructed with different settings of 'level'. We assume that the reviewer is referring to MCBS of level 1 only, because MCBS of higher levels has no resemblance to SGBS.
>
> &nbsp;
>
> **MCTS that can replace SGBS**
>
> Efficiency and scalability are serious issues for neural approachs to combinatorial optimization, and for a good reason. Because highly efficient heuristic methods already exist that are not learning-based for most of the popular benchmark CO problems, neural approaches are facing harsh challenges from the OR community for practicality on the basis of efficiency and scalability.
>
> When we deal with computationally expensive deep neural net models, MCTS is not too promising. Even with good parallelization techniques for MCTS, there are many levels of difficulties in order to make it work as efficiently as SGBS. Technically, first of all, it is not trivial to implement a parallel tree search running on GPUs. It is one thing to run a classical, CPU-based MCTS algorithm in parallel via multi-threading, but when it needs operations on GPUs, you would normally need multiple GPUs each matching with its partnered thread of CPU execution. For large scale projects, it is certainly doable, but economically it should not be a favored option.
>
> Second, there is a GPU memory issue to cope with. Neural net models contain so many parameters, and the trend is having more. The policy model for FFSP in our experiments, for example, requires a memory size of more than 100 MB.
>
> We have developed SGBS with meticulous attention to these efficiency issues. Our SGBS is optimized in its theory of operation and also in implementation as Python code. To traverse a node one step down in the search tree, SGBS needs just a single cycle of its Simulation phase. And during the Simulation phase, all rollouts are executed in parallel as one batch, from candidate child nodes to their termination nodes, in a single loop. MCTS can hardly match this level of parallelizability.
>
> Finally, let us briefly remind you of the difficulty of designing the MCTS just to have it perform properly with the policy models we use in our experiments. This problem is explained in our answers to Reviewer q7i4, under the heading 'Q3&Q4. Problem with MCTS applied on RL-trained models.' Despite the huge success with AlphaGo, MCTS with policy neural networks has not been a popular topic in the deep learning community, and there is a lack of study on its proper structure, the development of which may not be a trivial problem.

---

### Meta-Review · Area_Chair_CfaD · 2022-08-29

**Recommendation:** Accept
**Confidence:** Less certain

**Metareview:**

The paper follows in the footsteps of alpha go and presents two methods for neural-network guided search, targeting in particular beam search. The paper was deemed a bit incremental, but the method is simple, is easier to parallelize than MCTS and obtains good results on problems under-explored in machine learning. Please review related literature in AI for games and neural guided search techniques in discrete inference.


**Award:**

No

---

### Decision · Program_Chairs · 2022-09-14

Accept